# Exploration by Uncertainty in Reward Space

## Abstract

Efficient exploration plays a key role in reinforcement learning tasks. Commonly used dithering strategies, such as $\epsilon$-greedy, try to explore the action-state space randomly; this can lead to large demand for samples. In this paper, We propose an exploration method based on the uncertainty in reward space. There are two policies in this approach, the exploration policy is used for exploratory sampling in the environment, then the benchmark policy try to update by the data proven by the exploration policy. Benchmark policy is used to provide the uncertainty in reward space, e.g. td-error, which guides the exploration policy updating. We apply our method on two grid-world environments and four Atari games. Experiment results show that our method improves learning speed and have a better performance than baseline policies.

## 1 Introduction

Reinforcement learning (RL) methods aim at enabling agent to learn policies to maximize cumulative rewards from an unknown environment. Unlike traditional planning problems, the agent does not know perfect information of the environment, but needs to learn the information through experience. Therefore, how to get the necessary experience is a key issue in reinforcement learning tasks. Generally, agents need to learn through trial and error in the environment. Unless the agent fully explores the environment and identifies all opportunities for high rewards in all situations, it is impossible for the agent to take near-optimal action in the environment. Besides, this demand introduces a fundamental trade-off between exploration and exploitation as well. Generally, the agent may improve its future rewards by exploring states and actions which are not well understood. However, this exploration process may sacrifice immediate rewards of RL agent. To learn efficiently, RL agents should identify which states are worth exploring. Therefore, designing algorithms which can well trade-off between exploration and exploitation is urgent and crucial.

Generally, there are two main methods of exploration: exploration in state-action space and exploration in parameter space. State-action exploration Thrun & Möller (1991); Pathak et al. (2017) tries to systematically explore the state-action space, such as selects different action when state $S$ is visited. Parameter exploration Rückstieß et al. (2010) methods pick different parameters of policy $\pi$ and runs $\pi$ for a period of time. The advantage of parameter exploration is that it can take consistent exploration. However, the changes in parameter space can not can not directly reflect the change of action state space. RL agent may prefer a method which can not only intuitively represent the uncertainty of state action space, but also continuously explore.

A lot of policies fall into the category of exploration in state action space. For example, common dithering strategies such as $\epsilon$-greedy, approximate the value of an action with a number. With probability (1-$\epsilon$), this method picks the action with the highest estimate value, expected to get the best promising return. Otherwise, it picks one of the available actions at random. Recent work has considered scaling exploration strategies to large domains Bellemare et al. (2016). Several of these papers have focused on employing optimism-under-uncertainty approaches, which essentially rely on computing confidence bounds over different actions, and acting optimistically with respect to that uncertainty.

Recently, There are many simple but efficient exploration strategies, such as uniform sampling Mnih et al. (2015a) and correlated Gaussian noise strategies Schulman et al. (2015). These heuristics works well in tasks with well-shaped rewards. When the rewards in RL tasks is sparse, the sample complexity can grow exponentially with the state space size increasing Osband et al. (2016b). Recently

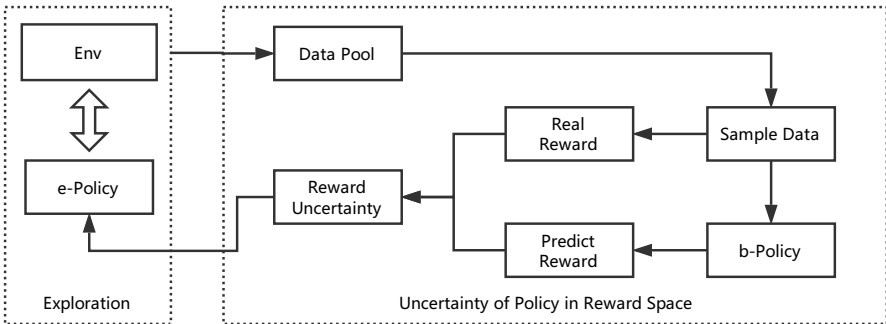

Figure 1: The e-Policy means exploration policy, which is used for interacting with the environment and get exploration experience into data pool. Then the framework samples data from experience data pool to update the benchmark policy (b-policy). According to the sample data, we can get the real reward which is given by the environment. On the other hand, the benchmark policy can also provide a predict reward. The difference between real reward and predict reward means the uncertainty of benchmark policy in reward space. Reward uncertainty is used to guide exploration strategy updating.

developed deep RL exploration strategies have significantly improved performance in sparse rewards environments. Bootstrapped DQN Osband et al. (2016a) trains an ensemble of Q-functions and thus gets faster learning in a range of Atari games than dithering strategies. Intrinsic motivation strategies using pseudo-counts and achieve state-of-the-art performance on Montezuma's Revenge Bellemare et al. (2016), which is an extremely challenging Atari game. In addition, Tang et al. (2017) proposes a generalization of classic count-based exploration on high dimensional spaces through hashing and have good effectiveness on challenging deep RL benchmark problems. Previous RL literature Osband et al. (2016a) provides a variety of provably and efficient approaches for exploration. However, most of them are limited to Markov decision processes (MDPs) with small and finite state spaces, which may not be suitable for complex environments in practice. Therefore, the statistical nature of most exploration strategies is not efficient in large-scale applications Mnih et al. (2015a).

In this paper, we propose a method to explore in reward space. This approach prefer to choose a state with large uncertainty in reward space, that is, if the reward estimate for a state is less accurate, we are more likely to explore that state. As shown in Figure 1, there are two policies in this approach, e-Policy means exploration policy, which is used for interacting with the environment and then get exploration experience into data pool. Then the framework samples data from experience data pool to update the benchmark policy (b-policy). According to the sample data, we can get the real reward (or n-step returns) which is given by the environment. On the other hand, the benchmark policy can also provide a predict reward (or n-step returns). The difference between real reward (or n-step returns) and predict reward (or n-step returns) means the uncertainty of benchmark policy in reward space. Reward uncertainty is used to guide exploration strategy updating.

The consequent sections respectively present the related works, describe the proposed approach, analyze the empirical results, and finally conclude the paper.

## 2 RELATED WORK

The trade-off between exploration and exploitation has been widely studied in previous reinforcement learning literature Kearns & Singh (2002); Strehl et al. (2009); Abeille & Lazaric (2017). Auer (2002) addresses this question for multi-armed bandit problems and provided regret guarantees. Jaksch et al. (2010) investigates the regret of the optimistic model in undiscounted reinforcement

learning processes. Bartók et al. (2014) studies a general case of partial monitoring games with finitely actions and outcomes, and provides minimax regret analysis.

Many exploration methods have been well studied in previous works. Generally, there are two main methods of exploration: exploration in state-action space Thrun & Möller (1991); Pathak et al. (2017) and exploration in parameter space Fortunato et al. (2017).

State-action exploration methods have been widely used in reinforcement learning tasks. One of the most familiar algorithm is $\epsilon$-greedy, with probability ($1$-$\epsilon$), this method pick the action with the highest estimate value to get the best promising return, otherwise, it picks one of the available actions at random. $\epsilon$-greedy provides a simple way of exploration in state-action space.

Uncertainty-driven search in state-space has also been advanced over the years Sutton (1990); Kolter & Ng (2009). In uncertainty-based exploration, if a state has not been studied sufficiently, then the state will have high uncertainty and the reinforcement learning agent will attend to explore these states with high probability. This ensures that an agent will thoroughly visit new areas of the state-action space. Classic count-based method such as Whitehead (1991) provides one of the earliest uncertainty-driven exploration policies. This work proposes a count-based rule that weighs each action based on the number of steps that have been taken since the last use of the action. Then actions are selected at random according to its weights. Besides, Kearns & Singh (2002) maintains a list of how many times each state has been visited, and try to explore the state with fewer visits. Strehl & Littman (2005) presents a theoretical analysis of model-based interval estimation and proves its efficiency even under worst-case conditions. Kolter & Ng (2009) presents a simple greedy approximation algorithm which is able to perform nearly as well as the optimal Bayesian policy after executing a several number of steps. In the methods mentioned above, agents can get rewards immediately throughout the state-action space. Recently, value-based Mnih et al. (2015b) and policy gradient-based Schulman et al. (2015); Liu et al. (2018) provide a crucial approach on the exploration of state-action space. These methods obtain rewards signal based on rollouts collected from interacting with environments and update the value function or policy parameters according to the rollouts in environment.

Another related categories of exploration methods is based on the idea of optimism in the face of uncertainty Brafman & Tennenholtz (2002); Osband & Roy (2014). These methods have rigorous theoretical guarantee in tabular settings. Besides, Bayesian reinforcement learning approaches study the distribution over MDPs Kolter & Ng (2009); Guez et al. (2014), and these methods are well extended to continuous space Pazis & Parr (2013); Osband et al. (2016b).

To deal with high-dimensional state space, Osband et al. (2016a) consider an alternative approach, named bootstrapped DQN, to exploration inspired by Thompson sampling. This method explores in the environment by randomly select a policy according to the probability it is the optimal policy. Houthooft et al. (2016) uses information gain about the agents internal belief of the dynamics model to drive the agent explore in the environment. In the two methods mentioned above, uncertainty of the model are considered when exploration. However, in this paper, we mainly focus on the uncertainty in the reward space. In our approach, if the reward estimate of a state is less accurate, we are more likely to explore that state.

Curiosity-based exploration methods provide a great idea of exploration Oudeyer & Kaplan (2009); Schmidhuber (2010). These methods try to get the surprise of the agent when interacting with the environment. The agents under curiosity-driven prefers to discover novel states in order to optimize the surprise. Pathak et al. (2017) generates an intrinsic reward signal based on the hardness of the agent to predict its own actions in the next. This work formulate curiosity as the error of an agents ability to predict the consequence of its actions in a new feature space learned by a self-supervised inverse dynamics model. Curiosity-driven approach tends to explore states that have not been seen before. But the state that has not been seen is not directly related to reward. Therefore, it may occur in a situation where a state has not been seen, but in the long run, it has little effect on reward. In this paper, we use the uncertainty in reward space as an intrinsic reward for policy exploration, directly associating reward with states, and exploring states that are more valuable and with higher reward uncertainty.

## 3 PROPOSED METHOD

A Markov decision process(MDP) can be defined by $(S, A, T, \gamma, r)$, where $S$ is the state space, $A$ is the action space, $T$ is a transition function, given state $s$ and action $a$, $T(s, a)$ defines the distribution of the next state. $r : S \times A \to R$ defines the reward function and $\gamma \in (0, 1]$ is a discount factor. In addition, given state $s$, policy $\pi(s)$ provides a distribution over actions. Reinforcement learning tasks try to learn an optimal policy $\pi*$ to maximize the total expected discounted reward $J(\pi^{\star}) = E_{\pi^{\star}}[\sum_{t=1}^{\inf} \gamma^{t-1} r_t]$

The main idea of the proposed method is that, when the reward of a state is not well learned and the uncertainty in reward space is large, then we should explore the state. There are two policies in our framework, exploration policy is used for interacting with the environment and then get exploration experience. Then the framework samples data from experience data pool to update the benchmark policy. According to the sample data, we can get the real reward which is given by the environment. On the other hand, the benchmark policy can also provide a predict reward. In this paper, the difference between real reward and predict reward is treated as the uncertainty of benchmark policy in reward space. Then exploration policy try to interact with the environment driven by reward uncertainty of benchmark policy. Under this settings, exploration policy try to visit the state which are not well studied in the reward space of benchmark policy.

As can be seen from Figure 1, the proposed method is an off-policy updating process. Benchmark policy are used to guide exploration policy visit states with high reward uncertainty in the environment. The connection between them is that both of them use the same data to train policy model. In this method, the exploration policy and the benchmark policy are separate. The benchmark policy does not directly interact with the environment. It only uses the data obtained by the exploration strategy in the environment to update itself. Then, the benchmark policy predicts the reward of a state, and the difference between predict reward and the reward obtained in the environment is taken as the uncertainty of the state in reward space. Then, in order to explore a more valuable state, the uncertainty in reward space is used as an intrinsic reward for the exploration strategy, guiding the exploration strategy to be further updated.

To illustrate this problem in more detail, in the next of this paper, we use the uncertainty in Q-value space instead, which better reflect long-term returns of state-action pairs. First, like the sampling process of DQN, eDQN samples from the environment, and selects the action with the largest Q value of exploration policy to execute. Note that we do not used $\epsilon$-greedy here. Given sample from the data pool, benchmark policy try to update according to the td-error in Q-learning. Then the mean square error of Q-value in benchmark policy is set as intrinsic reward of the exploration policy, and motivate it to perform higher Q-value of state-action pairs. The total process of deep Q-learning with exploration by reward uncertainty (eDQN) is shown in Algorithm 1.

In Algorithm 1, the data pool (replay memory) is set with capacity $N$, state-action function $Q$ is the Q-function of benchmark policy, $\hat{Q}$ is the Q value function of exploration policy. Given state $s_t$, eDQN select the action with maximum $\hat{Q}$ values with no $\epsilon$-greedy, where $a_t = \max_a \hat{Q}(\phi(s_t), a; \theta)$, $\phi(s_t)$ processes state data and generates new features. Then we can get the next state from environment $s_{t=1} = s_t, a_t$ and preprocess $\phi_{t+1} = \phi(s_{t+1})$. In addition, the Immediate reward $r_i$ can also be obtained from the environment. Then we can get a transition $(\phi_t, a_t, r_t, \phi_{t+1})$ in the environment and store it into replay memory $D$. In eDQN, we can sample a mini batch of transitions from $D$, and compute the target Q value from Q-learning, which can be written as

$$y_j = \begin{cases} r_j & \text{for terminal } \phi_{j+1} \\ \mathrm{r}_j + \gamma \max_{a'} Q(\phi_{j+1}, a'; \theta) & \text{else} \end{cases}$$

According to mini bath sample from $D$ and $Q$ function of benchmark policy, the mean square error of Q value is $L_j = (y_j - Q(\phi_{j+1}, a'; \theta))^2$. Then $L_j$ is regarded as the intrinsic reward of exploration policy, which shows the uncertainty of the benchmark strategy on $Q(s, a)$. Then the reward of $\hat{r}_j$ can be written as $\hat{r}_j = r_j + L_j$. Then the target value of $\hat{Q}$ can be written as

$$\hat{y}_j = \begin{cases} \hat{r}_j & \text{for terminal } \phi_{j+1} \\ \hat{r}_j + \gamma \max_{a'} \hat{Q}(\phi_{j+1}, a'; \theta) & \text{else} \end{cases}$$

---

**Algorithm 1** Deep Q-learning with Exploration by Reward Uncertainty (eDQN)

---
**Require:**
    Initialize replay memory $D$ to capacity N
    Initialize action-value function $Q, \hat{Q}$ with random weights
**Ensure:**
 1: **for** episode = 1,$M$ **do**
 2:    Initialise sequence $s_1$ and preprocessed sequence $\phi_1 = \phi(s_1)$
 3:    **for** t = 1,$T$ **do**
 4:        Select $a_t = \max_a \hat{Q}(\phi(s_t), a; \theta)$
 5:        Execute action $a_t$ and observe reward $r_t$ and state $x_{t+1}$
 6:        Set$s_{t=1} = s_t, a_t$ and preprocess $\phi_{t+1} = \phi(s_{t+1})$
 7:        Store transition$(\phi_t, a_t, r_t, \phi_{t+1})$ in $D$
 8:        Sample random mini batch of transitions$(\phi_j, a_j, r_j, \phi_{j+1})$ from $D$
 9:

$$y_j = \begin{cases} r_j & \text{for terminal } \phi_{j+1} \\ \text{r}_j + \gamma \max_{a'} Q(\phi_{j+1}, a'; \theta) & \text{else} \end{cases}$$

10:        $L_j = (y_j - Q(\phi_{j+1}, a'; \theta))^2$
11:        $\hat{r_j} = r_j + L_j$
12:

$$\hat{y}_j = \begin{cases} \hat{r_j} & \text{for terminal } \phi_{j+1} \\ \hat{r_j} + \gamma \max_{a'} \hat{Q}(\phi_{j+1}, a'; \theta) & \text{else} \end{cases}$$

13:        $\hat{L}_j = (\hat{y}_j - \hat{Q}(\phi_{j+1}, a'; \theta))^2$
14:        Perform a gradient descent step on $L_j$ update Q
15:        Perform a gradient descent step on $\hat{L}_j$ update $\hat{Q}$
16:    **end for**
17: **end for**

---

and the error of $\hat{y}_j$ is $\hat{L}_j = (\hat{y}_j - \hat{Q}(\phi_{j+1}, a'; \theta))^2$. According to $L_j$ and $\hat{L}_j$, state-action value function $Q$ and $\hat{Q}$ can be updated. $Q$ value function try to get a good policy from data pool while $\hat{Q}$ value function try to explore the states where $Q$ function does not learn very well. The more incorrect the Q value is, the more likely the state-action pair is to be explored. However, $\hat{Q}$ does not always perform state action pairs with higher uncertainty in Q-value, it is also affected by immediate reward $r$ of next state. That is, if the immediate reward of next state is relatively high, and its uncertainty of Q-value in benchmark policy in this state action pair is relatively high as well, then the total reward, where total reward = external reward + intrinsic reward, of exploration policy in this state will be high. Therefore, the exploration policy is more inclined to explore this state.

One thing to note is that there are multiple sampling methods when sampling from the dataset. One way to speed up the update is to use priority replay Schaul et al. (2016). More specifically, we sample transitions from the replay buffer using $(y_j - Q(\phi_{j+1}, a'; \theta))^2$ as the priority. This naturally increases the proportion of valid samples to contribute to the gradient. Besides, in order to make the DQN update strategy more stable, we also use the target network to assist the policy update.

It can be seen from the algorithm that as long as the benchmark estimates the Q value and the target Q value is biased, eDQN can continue to explore. Moreover, since the previous benchmark policy has not yet been learned well, the exploration may be more dependent on the intrinsic reward, that is, the uncertainty of Q value. In the later stage, when the benchmark tactics are better, the benchmark strategy estimates that the Q value is more accurate, the uncertainty is reduced, and the execution of the eDQN is more dependent on the external reward.

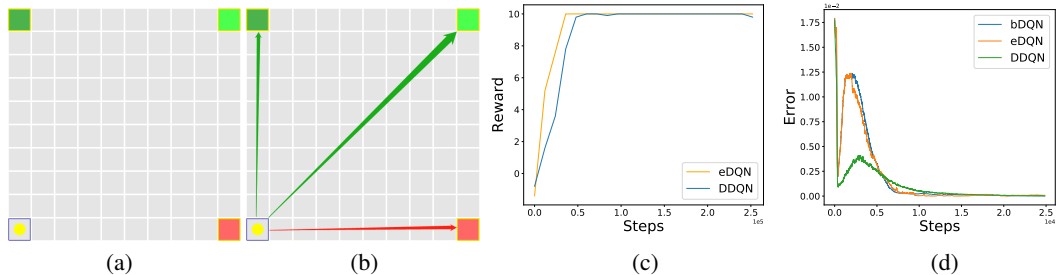

(a)            (b)            (c)            (d)

Figure 2: Grid world environment and experimental results. (a) shows the environment setting of grid world, reinforcement learning agent start from the yellow point at the lower left corner. The red and green grids represent the termination grid, where the red grid indicates that the reward is negative and the green grid indicates that the reward is positive.(b) shows the three optimal paths from the starting point to the ending states. (c) shows the rewards of eDQN and DQN in the environment as the steps increases. (d) shows the change in the loss value of each policy as the number of steps increases, where bDQN means the benchmark policy in the proposed frame work.

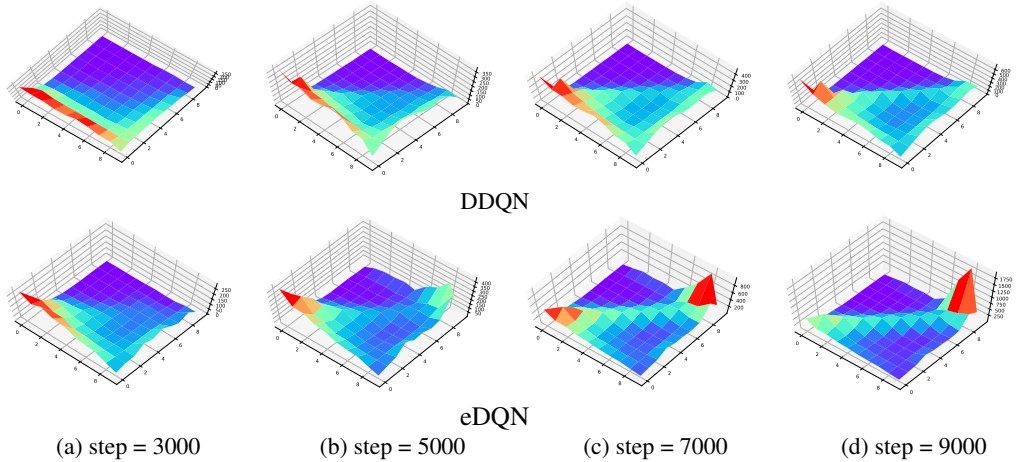

(a) step = 3000      (b) step = 5000      (c) step = 7000      (d) step = 9000

Figure 3: State distribution of DDQN and eDQN at step of 3000-5000-7000-9000. The brighter the color of the grid, the more times the grid is accessed. Red represents a large number of visits. Cyan and green indicate moderate number of visits. Blue indicates a small number of visits.

## 4 EXPERIMENT

We validate our approach on two grid world environments and four reinforcement learning tasks. The benchmark policy update method is DDQN van Hasselt et al. (2016) with $\epsilon$-greedy.

### 4.1 EXPLORATION ON GRID WORLD ENVIRONMENT

We first test our exploration methods in simple grid world environments. The first grid world environment size is $10 \times 10$. As can be seen in Figure 2. The agent starts from the lower left corner and has 8 directions that can move freely. In the grid world, it has three termination states can be reached, where the green grid indicates that the reward is positive and the red grid indicates that the reward is negative. In addition, The state in the upper right corner has the largest reward, set to 10, and the reward in the upper left corner state is set to 2, the lower left state of the reward is set to -5. Except for the three termination states, the immediate reward of other grid is set to 0. Detailed grid world environment as shown in 2 (a),(b). In our experiments, the policy network was a two-layer fully connected network of $128 \times 128$.The state space is the coordinates of the agent in the grid world, where the coordinates of the starting point are $(0, 0)$ and the coordinates of the three ending states are $(0, 9)$, $(9, 0)$, $(9, 9)$, respectively. The action of the agent is a value of 1-8, where each value represents a direction, that is, the agent can move in 8 directions in the grid world.

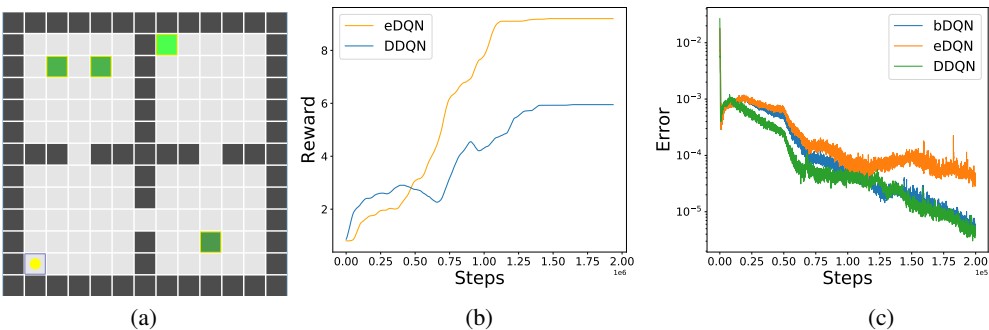

Figure 4: (a) shows the room environment of grid world, the black grid is an obstacle and cannot pass, the dark green grid reward is 1, and the light green grid reward is 10. (b) shows the reward curves of eDQN and DQN (c) shows the loss curve of DDQN, bDQN, eDQN

Figure 2 (c) shows the reward comparison of eDQN and DDQN, as can be seen from Figure (c), both eDQN and DQN quickly converge to the optimal solution, and eDQN converges faster than DDQN. Figure 2 (d) shows the change of loss in each method as the number of iterations increases, where bDQN is the benchmark DQN. The loss of bDQN is the intrinsic reward of eDQN. It can be seen from (d) that the bDQN loss is large in the early stage and the bDQN loss becomes smaller in the later stage. In the current reward sparse grid world, eDQN is affected by the uncertainty of Q-value in the early stage. When the iteration steps are greater than 7k, the loss of bDQN decreases and tends to be stable. At this point, eDQN is less affected by uncertainty. In addition, figure (d) shows that the loss convergence of eDQN and bDQN is faster than DDQN, which verifies the effect of Figure c on the other hand.

Figure 3 (a),(b),(c),(d) show the state distribution of DDQN and eDQN at 3000, 5000, 7000, 9000 steps respectively. As can be seen from the figure, at 3000 and 5000 steps, eDQN explores more states than DDQN, and at 7000 steps, eDQN finds the maximum target state of reward, and has a large number of trajectory states near this state. At 9000 steps, eDQN learned the optimal path, while the path of DDQN does not as well as eDQN.

To further investigate the differences between eDQN and DQN, we compared the two methods in a more complex grid world environment. Figure 4 (a) shows the four-room environment with four termination grids, a dark green grid indicating a reward of 1, and a light green grid representing a reward of 10. The agent starts from the lower left corner and has 8 directions for each step. It stops walking after reaching a termination state and gets a reward. It can be seen that this is an environment with three local optimal solutions. Figure 4 (b) shows the effect of DDQN and eDQN on the environment. It can be seen that eDQN converges faster and obtains a better solution than DDQN. (c) shows the loss curve of DDQN, bDQN, and eDQN. It can be seen that the loss of bDQN is gradually decreasing, that is to say, the influence of intrinsic reward on eDQN is gradually diminishing in the later period, and the reward of eDQN is more concerned about the reward of environment in the later period.

## 4.2 EXPLORATION ON ATARI GAMES

Atari video games provide an important benchmark for deep reinforcement learning due to its high-dimensional state spaces and wide variety of games. To further validate the effects of eDQN, we selected four Atari game environments, SpaceInvaders, BreakOut, Enduro and Pong, for our experiments. In environments SpaceInvaders and Pong, an appropriate exploration will accelerate the convergence of the policy and get a better solution, improper exploration will bring some losses, so the two environments are more sensitive to the exploration strategy. In BreakOut and Pong environment, agent is free to explore for a period of time before it finally gets to the target, and the actions it does during that time do not have much impact on the final result (the agent only needs to choose the right action in the last few steps to get the final reward). Therefore, the agent may be less affected by the exploration in the breakout environment. In addition, we used DDQN and noisy DQN Fortunato et al. (2017) as comparison methods, where DDQN uses $\epsilon$-greedy as a exploration strategy. Besides,

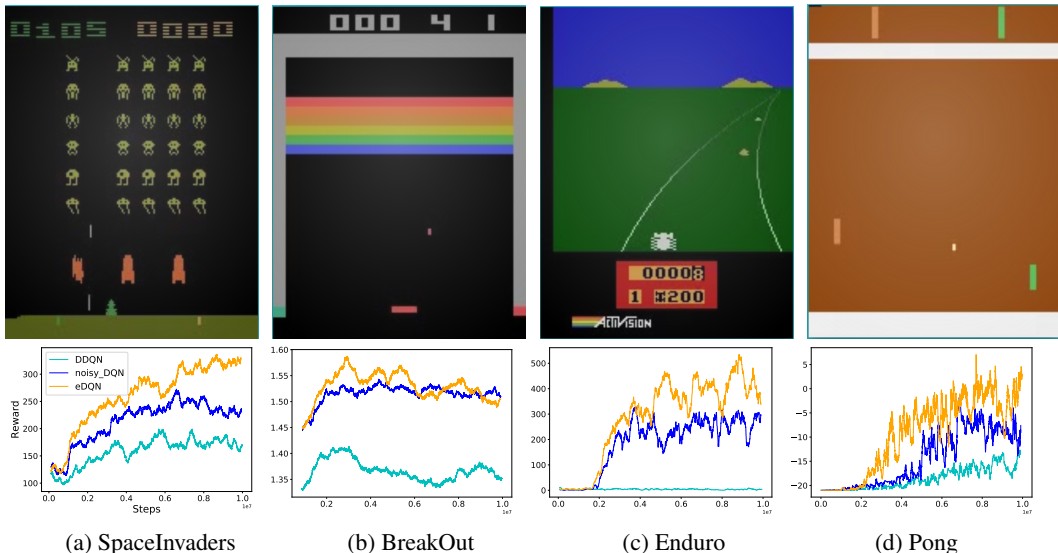

Figure 5: Performance of the method with the number of samples; (a), (b), (c), (d) are performances comparison of DDQN, noisy DQN, eDQN in terms of the average reward on the SpaceInvaders task, BreakOut task, Enduro task and Pong task, respectively.

noisy DQN is a deep reinforcement learning method with parametric noise added to its weights, which try to explore in the parameter space and can be used to get efficient exploration. Policies and value functions are neural networks with the same architectures as Dhariwal et al. (2017). Besides, all three methods use target network assisted training, and use the priority method for sampling, speeding up policy updates.

The top four images in Figure 5 correspond to four environments mentioned above. The following four figures are the reward curves of eDQN, DDQN and Noisy DQN in the four environments. In each environment, we executed 10 million steps in the environment and plotted its reward curve as the number of steps increased. As shown in Figure 5 (a) and (c), eDQN achieved the best performance in the SpaceInvader and Enduro environments, and in these three environments, Noisy DQN performed better than DDQN with $\epsilon$-greedy. It shows that in the two environments that are sensitive to exploration strategies, eDQN is better than Noisy DQN to some extent. Figure 5 (b) and (d) shows that DDQN learned a relatively poor strategy in 10 million iterations on BreakOut and Pong. In the two environments which are not sensitive to exploration strategies, eDQN and Noisy DQN perform similarly. In BreakOut environment, eDQN gets better values in the early stage, and Noisy DQN is very close to eDQN in the later stage. In Pong, eDQN perfroms better than Noisy DQN. However, the gap between the two strategies is not obvious.

In general, eDQN is better than noisy DQN and DDQN with $\epsilon$-greedy in environments that are sensitive to exploration. For exploring insensitive environments, eDQN and noisy DQN perform better than DDQN with $\epsilon$-greedy. In addition, Compared with noisy DQN, eDQN performs somewhat well. but the difference between eDQN and Noisy DQN is not obvious.

## 5 CONCLUSION

In this paper, We propose an exploration method based on the uncertainty in reward space. In this approach, there are two policies, named exploration policy (e-policy) and benchmark policy (b-policy). The exploration policy is used for exploratory sampling in the environment and get data into data pool, the benchmark policy uses samples from data pool to update itself, and predicts the rewards (immediate reward or n-step return) of that these states. Then the uncertainty in reward space, e.g. error between predicted reward and real reward, of benchmark policy is used as an intrinsic reward for the exploration policy. We apply our method on two grid-world environments and four Atari games.

Experiment results show that our method improves learning speed and have a better performance than baseline policies.

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
