# OpenReview forum: "Exploration by Uncertainty in Reward Space"
_ICLR.cc/2019/Conference_

### Official Review · AnonReviewer2 · 2018-10-30
**A well-intentioned piece of work... but the understanding of prior work / the exploration problem is lacking**

**Rating:** 3
**Confidence:** 5

**Review:**

This paper suggests an exploration driven by uncertainty in the reward space.
In this way, the agent receives a bonus based on its squared error in reward estimation.
The resultant algorithm is then employed with DQN, where it outperforms an e-greedy baseline.

There are several things to like about this paper:
- The idea of exploration wrt uncertainty in the reward is good (well... actually it feels like it's uncertainty in *value* that is important).
- The resultant algorithm, which gives bonus to poorly-estimated rewards is a good idea.
- The algorithm does appear to outperform the basic DQN baseline on their experiments.

Unfortunately, there are several places where the paper falls down:
- The authors wrongly present prior work on efficient exploration as "exploration in state space" ... rather than "reward space"... in fact prior work on provably-efficient exploration is dominated by "exploration in value space"... and actually I think this is the one that makes sense. When you look at the analysis for something like UCRL2 this is clear, the reason we give bonus on rewards/transitions is to provide optimistic bounds on the *value function*... now for tabular methods this often degenerates to "counts" but for analysis with generalization this is not the case: https://arxiv.org/abs/1403.3741, https://arxiv.org/abs/1406.1853

- So this algorithm falls into a pretty common trope of algorithms of "exploration bonus" / UCB, except this time it is on the squared error of rewards (why not take the square root of this, so at least the algorithm is scale-invariant??)

- Once you do start looking at taking a square root as suggested (and incorporating a notion of transition uncertainty) I think this algorithm starts to fall back on something a lot more like lin-UCB *or* the sort of bonus that is naturally introduced by Thompson (posterior) sampling... for an extension of this type of idea to value-based learning maybe look at the line of work around "randomized value functions"

- I don't think the experimental results are particularly illuminating when comparing this method to other alternatives for exploration. It might be helpful to distill the concept to simpler settings where the superiority of this method can be clearly demonstrated.

Overall, I do like the idea behind this paper... I just think that it's not fully thought through... and that actually there is better prior work in this area.
It could be that I am wrong, but in this case I think the authors need to include a comparison to existing work in the area that suggests "exploration by uncertainty in value space"... e.g. "deep exploration via randomized value functions"

---

### Official Review · AnonReviewer1 · 2018-10-30
**Nice paper**

**Rating:** 5
**Confidence:** 2

**Review:**

The authors develop a new algorithm for reinforcement learning based on adding another agent rewarded by both the extrinsic environment reward and the TD-errors of the original agent, and use the state-action pairs visited by the added agent as data for the original.

This co-evolutionary process could be more broadly described as a student agent that learns from the trajectories of a teacher agent, which visits trajectories with high reward and/or high student TD-error.

The algorithm is not proved to converge to the optimal Q.
Algorithm (1) by not using epsilon-greedy on Q_hat
has an initialization-based counter-example in the tabular bandit case
e.g.
  MDP being a bandit with two arms with rewards 100 and 1000 respectively
  Q_hat that initially is X for first arm and Y for second arm, with X > 100 > Y
This could be solved by, for example, adopting epsilon greedy.

Prioritized Experience Replay (Schaul et al. https://arxiv.org/pdf/1511.05952.pdf), which suggests using the TD-error to change the data distribution for Q-learning, should be also a baseline to evaluate against.
[Speculative: It feels like a way to make prioritized experience replay that instead of being based on time-steps (state, action, reward, new state) is based on trajectories, and this is done by doubling the number of model parameters.]

On quality:
The evaluation needs different experience replay baselines.

On clarity/naming:
Here 'uncertainty in reward space' is used to refer to TD-error (temporal difference), I found that confusing.
Here 'intrinsic motivation' is used but the 'intrinsic motivation' proposed depends on already having an externally defined reward.

Pros:
+ "Prioritized Experience Replay for Trajectories with Learning"
Cons:
- Not evaluated against experience replay methods.
- No plots showing number of gradient descent steps (as the proposed method has double gradient descent updates than the baselines)
- No proof of correctness (nor regret bounds).

---

### Official Review · AnonReviewer3 · 2018-11-07
**An interesting form of "imitation learning".**

**Rating:** 5
**Confidence:** 3

**Review:**

This paper considered the idea of accelerating sampling process by exploring uncertainty of rewards. The authors claimed more efficient sampling by building a reference policy and an exploration policy. Algorithm was tested on grids and Atari games.

The authors proposed eDQN, which is more like exploring the uncertainty of Q values instead of rewards. The reviewer is also expecting to see the convergence guarantee of eDQN.

The paper is well-organized and easy to understand. Written errors didn't influence understanding.

---

### Meta-Review · Area_Chair1 · 2018-12-14
**Nice ideas, but need a better comparison to previous work**

**Confidence:** 4
**Recommendation:** Reject

**Metareview:**

The paper has some nice ideas for efficient exploration, but reviewers think more work is needed before it is ready for publication.  In particular, the paper should have an improved discussion of state-of-the-art work on exploration, compare the difference and benefits of the proposed approach, and then conduct proper experiments to validate the claims.